# Accounting for drinking water quality in measuring multidimensional poverty in Ethiopia

**Alemayehu Azeze Ambel**[1]*, **Harriet Kasidi Mugera**[2], **Robert E. S. Bain**[3]

**1** Development Data Group, World Bank, Washington, DC, United States of America, **2** Development Data Group, World Bank, Copenhagen, Denmark, **3** Division of Data, Analysis Planning and Monitoring, United Nations Children's Fund (UNICEF), New York, NY, United States of America

* aambel@worldbank.org

## Abstract

The Multidimensional Poverty Index is used increasingly to measure poverty in developing countries. The index is constructed using selected indicators that cover health, education, and living standards dimensions. The accuracy of this tool, however, depends on how each indicator is measured. This study explores the effect of accounting for water quality in multidimensional poverty measurement. Access to drinking water is traditionally measured by water source types. The study uses a more comprehensive measure, access to safely managed drinking water services, which are free from *E. coli* contamination, available when needed and accessible on premises in line with Sustainable Development Goal target 6.1. The study finds that the new measure increases national multidimensional headcount poverty by 5–13 percentage points, which would mean that 5–13 million more people are multidimensionally poor. It also increases the poverty level in urban areas to a greater extent than in rural areas. The finding is robust to changes in water contamination risk levels and Multidimensional Poverty Index aggregation approaches and weighting structures.

## 1. Introduction

Measuring poverty is a contentious issue in the development literature. Using income or consumption spending as a proxy for household economic welfare is a common way to estimate poverty in developing countries [1], but the approach has practical, empirical, and theoretical limitations, which Alkire and Santos [2] have summarized. These two indicators often fail to capture other crucial dimensions of poverty especially in developing countries. For instance, people who are consumption poor are nearly the same as those who suffer malnutrition, are ill-educated, or are disempowered. Data availability can be a practical constraint because income and consumption surveys are costly, time-consuming, and complex; and it is difficult to find quality data at adequate frequency for every country. Lack of comparable consumption and spending data also limits international comparisons because different countries conduct such surveys differently and in different frequencies.

**Data Availability Statement:** All relevant data are available online at World Bank Microdata Library: https://microdata.worldbank.org/index.php/catalog/2783

**Funding:** The authors did not receive any specific funding for this study. This task is done as part of a regular work. Authors received salaries from their respective institutions. AA and HM from World Bank and RB from UNICEF.The funders had no role in study design, data collection and analysis, decision to publish, or preparation of the manuscript.

**Competing interests:** The authors have declared that no competing interests exist.

The poverty line derived using income and consumption spending may sometimes be inaccurate due to differences in the prices people in different places pay. It may also be hard to ensure that people's minimum needs are being met because of differences in consumption behaviour [3]. Moreover, income or consumption-based measures of poverty do not provide much policy guidance regarding deprivations in these and other dimensions.

The concept and methodology for identifying dimensional poverty tackles some of the limitations of standard poverty measures that are solely monetary. This approach is now widely used to monitor national and global development goals. For example, one Sustainable Development Goal (SDG) target is by 2030 "to reduce at least by half the proportion of men, women and children of all ages living in poverty in all its dimensions according to national definitions." [4].

Currently, the most widely used measure of dimensional poverty is the Multidimensional Poverty Index (MPI) developed by the Oxford Poverty and Human Development Initiative (OPHI) [5]. The MPI approach to measuring poverty has certain desirable properties: (1) The variety of dimensions and indicators enables it to be adapted to different contexts and purposes. (2) The methodology can also be used to examine one sector, to represent, e.g., the quality of education or dimensions of health. (3) Ordinal, categorical, and cardinal data can be used. (4) The approach can be broken down into its individual dimensions to identify which deprivations are driving multidimensional poverty in different regions or groups. (5) Finally, it has the power to guide policy-making that efficiently addresses deprivations in different groups. It is also an effective tool for identifying priority areas for programmatic and policy interventions.

However, how effective the MPI is depends on the quality of the indicators that measure the dimensions [6]—the accuracy of the MPI metric depends on how each indicator was captured. In addition, for the MPI to measure functionings in the sense of Sen's capability approach, measuring only access to resources will not be enough [7]. Had it not been for data availability, some of the existing MPI indicators could be better captured by relevant and objective measures. For example, in education, literacy and numeracy test results are better measures of learning outcomes than enrolment status because the latter does not guarantee that students actually acquire the human capital that their schooling should provide [8]. Similarly, a mere connection to the national grid does not fully measure household access to energy. In measuring energy poverty, safety and sufficiency concerns need to be considered [9].

In this study, we reexamine the MPI for Ethiopia by changing how access to "safe" drinking water is measured. Access to safe drinking water is one of the ten MPI indicators and one of the six in the living standards dimension. Although different studies aggregate MPI in different ways using different indicators, the access to safe drinking water indicator is often included in the list. However, Dotter and Klasen [10] note that while this indicator is relatively easy to measure, it is based on water source type, not on whether the water is actually "clean".

Taking advantage of the availability of recent survey data on drinking water quality in Ethiopia, we adjust the water indicator. Instead of using the traditional classification of sources as improved and unimproved, we adjusted the improved indicator to take water quality, accessibility and availability into account [11]. In our analysis we define access to safely managed drinking water services as use of an improved source which is free from *E. coli* contamination, available when needed and accessible on premises [11]. A household is therefore considered deprived if it does not have access to safely managed drinking water services. We then generate a new MPI using this indicator and compare the resultant poverty rates with the baseline indicators in different scenarios at point of collection and at point of use. Depending on the estimated model and scope of analysis, we find that replacing the traditional measure of water

indicator with a more objective measure of quality pushes up multidimensional poverty (MDP) headcount by 5–13 percentage points (pp), which implies 5–13 million more MPI-poor individuals. The increase is observed in all regions. Our finding is also robust to changes in *E. coli* contamination risk levels and different MPI aggregation and weighting structures.

## 2. Materials and methods

### 2.1 Study setting

**2.1.1. Access to drinking water.** The results of the 2016 Ethiopia Water Quality Survey show that approximately 66 percent of the population accessed drinking water from improved sources such as piped water and protected wells and springs [12]. The remaining 34 percent of the population fetched their drinking water from unimproved sources such as unprotected springs and rivers. Access to improved sources of drinking water varies by location of residence, region and poverty levels. For example, in rural areas the rate was about 59 percent compared to 94 percent in urban areas. By region. almost all residents of the capital Addis Ababa and approximately 72 percent of residents in Tigray region reported using improved water sources compared to the national average of 66 percent. There were also substantial differences in access by poverty levels; about 55 percent of the population in the poorest quintile compared to about 82 percent in the richest quintile.

However, improved sources are not necessarily safe. The next level of access to safe drinking water considers biological and chemical contaminations. In this measure, only 13 percent of the population was considered to have access to safely managed drinking water. There is a substantial difference between safely managed and improved measures. At national level, the gap between the two measures is 53 percentage points. While access to safely managed drinking water services is in general very low, the rate is different for different groups. Approximately 5 percent of the population in rural areas and 37 percent in urban areas reported access to safely managed drinking water services. Similarly, regional variations range from approximately 7 percent in the SNNP region to 51 percent in Addis Ababa. Likewise, by poverty levels, access to safely managed drinking water services was about 4 percent in the poorest quintile compared to about 32 percent in the richest quintile.

**2.1.2 Multidimensional poverty and access to drinking water.** Recent studies have found that multidimensional poverty is very high in Ethiopia [13–16]. According to the Global MPI 2019 report, in 2016 about 83.5 percent of Ethiopians were multidimensionally poor, making Ethiopia the fifth poorest country in the world, above only Burkina Faso, Chad, Niger, and South Sudan [14]. The global report also shows substantial differences by place of residence and region. About 9 out of 10 individuals in rural areas are multidimensionally poor compared to less than 4 out of 10 in urban areas. Moreover, in the Ethiopian MPI most of the deprivation comes from living standards indicators, such as cooking fuel, sanitation, drinking water, electricity, housing, and assets. This pattern for Ethiopia has been documented in other studies using different data [13,17]. Another important observation is that the living standards deprivations are in general persistent over the long term. However, the water indicator has been found to be a driver of movement out of dimensional poverty [15,18]. However, as reported above there is a huge gap in safety between self-reported and direct measurement: by self-report (use of improved drinking water sources), deprivation is about 34 percent. Yet adjusting for direct measurement, based on *E. coli* contamination, deprivation shoots up to 84 percent at water source and 93 percent at point of use. This raises questions about the progress in MPI reduction over the years that was driven by increased access to water from improved sources. The study thus reexamines MPI in Ethiopia by applying alternative indicators for drinking water.

## 2.2 Data on drinking water services and their quality

We use data from the 2016 Ethiopia Water Quality Survey. The survey was implemented in the 2015/16 Ethiopia Socioeconomic Survey (ESS) sample households. It was conducted May–July 2016 [19]. Drinking water samples were collected from 4,688 households and 4,533 source points. The overall sample is representative of households in rural, small town, and medium and large towns in Ethiopia and six major regions. It is also representative for five major regions (Addis Ababa, Amhara, Oromiya, SNNP, and Tigray) and a sixth "region" comprising all other regions [12].

Figs 1 and 2 map drinking water source types by category (improved vs. non-improved) and *E. coli* contamination status at source and point of use. Fig 1 clearly shows that not all improved sources are safe. In fact, most of the improved sources are mapped to the contaminated-at-source category. Of the 4,444 households, 3,010 (68%) were getting water from improved sources. However, the test at source level found only 664 households (15% of the total sample) were free of *E. coli*. At point of use, this declines to 281 households (6 percent of the total sample) due to further contamination during transport and storage.

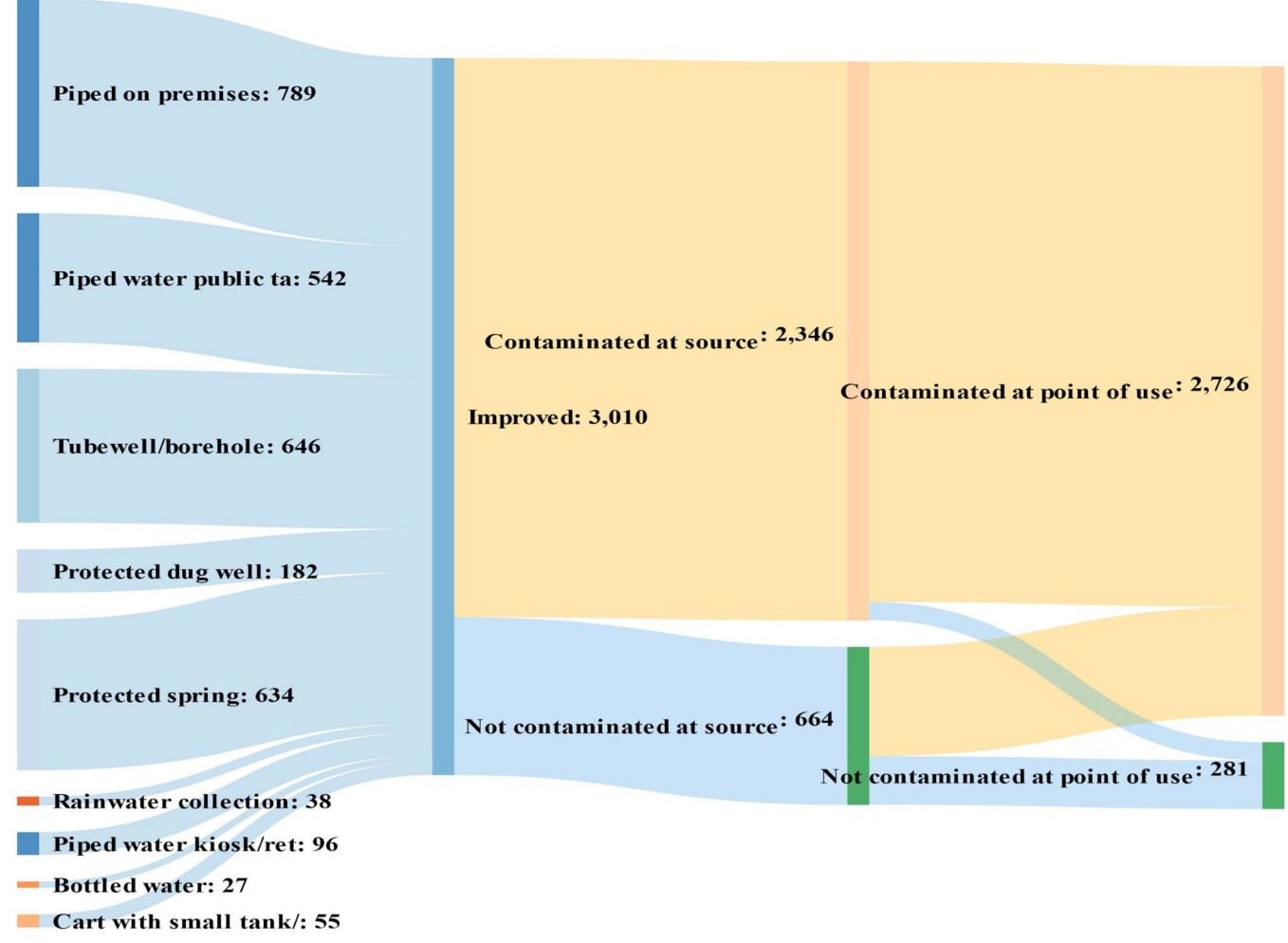

**Fig 1. Schematic diagram of improved water source types and their *E. coli* contamination status at source and point of use.** Authors' calculations based on ESS 2016.

Fig 2 is for unimproved sources, such as unprotected springs, unprotected dug wells, and surface water. As expected, almost all samples of unimproved water were found to be contaminated at both source and point of use. The self-report measure almost correctly classifies all unimproved sources as not free from contamination. However, it misclassified substantial number of contaminated sources as improved in both rural and urban areas (See S1 and S2 Figs). In rural areas, almost all water samples collected from both improved and unimproved sources were found to be contaminated at point of use.

The figures show that how measuring wellbeing could be affected differently by different measures of access to drinking water. This study explores this measurement effect using three indicators of lack access to drinking water within the MPI:

- *Drinking water indicator 1*: No improved source. This measure depends on the self-reported response that was provided by the household on the type of water source used for drinking. The head of the household or any other most eligible member of the household was the respondent to these questions as they are believed to be the most informed members of the household on water related issues in the household. Water sources were classified as improved or unimproved according to the definitions of the WHO/UNICEF JMP [20].

- *Drinking water indicator 2*: No safely managed source. This measure combines self-reported information on the type of water source, location and availability with direct measurement of water quality. Households were considered to lack a safely managed drinking water source if they used an unimproved drinking water source or relied on an improved source of drinking water which was located off premises, did not provide sufficient drinking water in the preceding month, or from which a measure of fecal contamination ($\geq$1 *E. coli* in a 100 ml sample) was detected in water quality samples collected from the source. This corresponds to the new SDG benchmark for drinking water services as monitored by WHO/UNICEF JMP.

- *Drinking water indicator 3*: No safely managed source free from contamination at point of use. As above but also considering water quality at the point of use. In addition to the criteria for safely managed drinking water services, households were considered to be deprived if *E. coli* were detected in the sample of drinking water at the point of use. This is a more stringent standard than used by WHO/UNICEF JMP as it reflects the quality of drinking water at the point of consumption.

## 2.3 The MPI methodology

In the MPI approach, poverty is measured in two distinct steps. First is the identification step—defining the cut-offs for distinguishing the poor from the nonpoor. If the person is poor, the identification function has a value of 1; if the person is not poor, the function has a value of 0. Second is the aggregation step—unifying data on the poor into an overall indicator of poverty. The identification stage has dual cut-offs: The first is the traditional dimension-specific poverty line. This cut-off, which is set for each dimension, identifies whether a person is deprived with respect to that dimension. The second cut-off describes how severely deprived a person must be to be considered poor. A poverty cut-off $k$ satisfying $0 < k \leq d$ is used to determine whether a person has enough deprivations, $d$, to be considered poor. The OPHI traditionally uses a cutoff of $k> = 0.33$ as the poverty threshold. Following the AF MPI methodology, MPI is calculated by multiplying the incidence of poverty (H) and the intensity of poverty (A). The incidence (H) captures the proportion of people who are multidimensionally poor and the intensity (A) represents the average proportion of dimensions in which poor people are deprived.

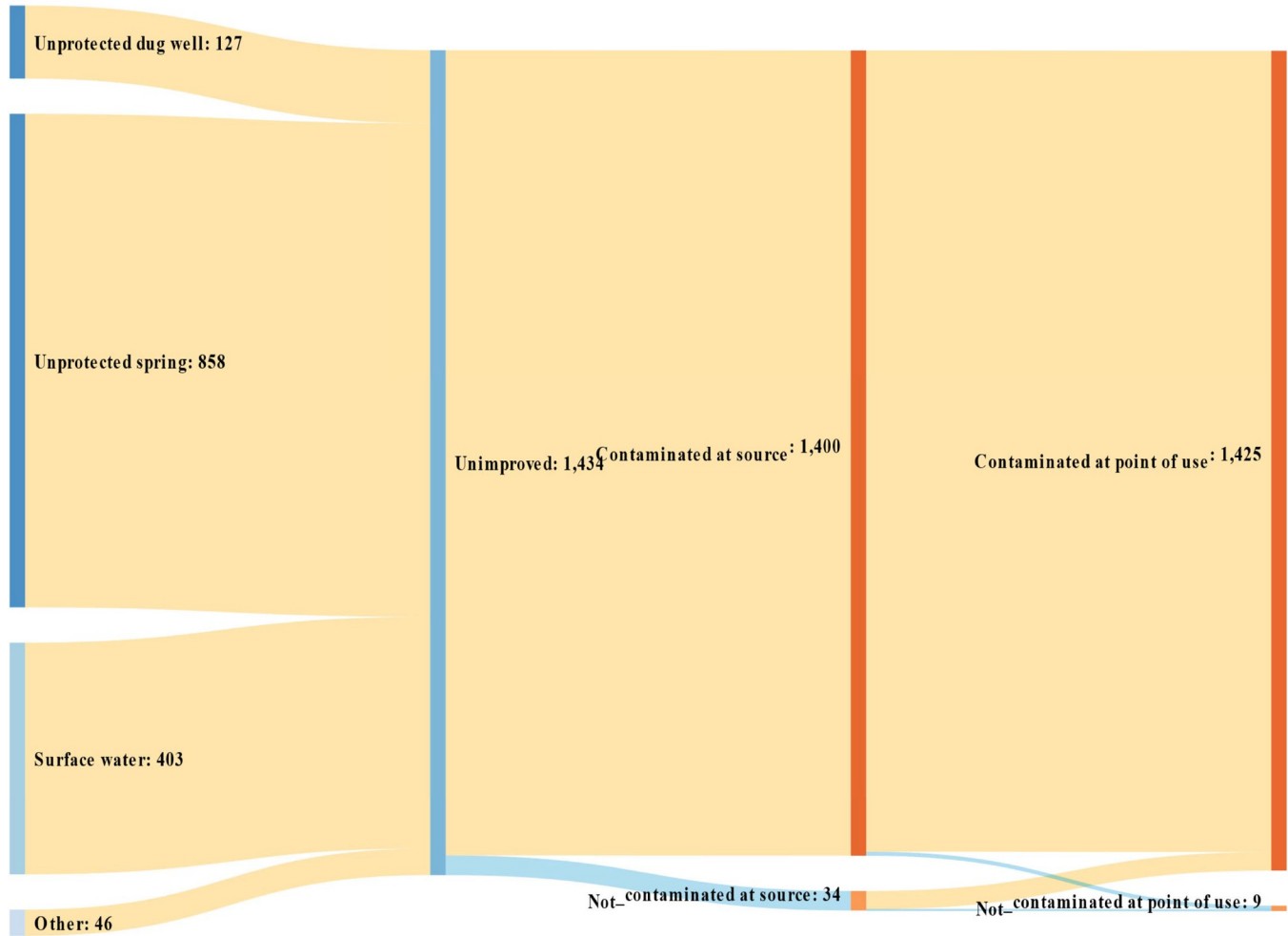

**Fig 2. Unimproved water source types and their *E. coli* contamination status at source and point of use.** Authors' calculations based on ESS 2016.

MDP is calculated by aggregating the indicators presented in Table 1. These indicators are the most used indicators in multidimensional poverty analysis. They are adapted from Alkire et al. [21]. While different dimensional poverty studies included different indicators and dimensions, access to drinking water is often included in the MPI calculation. Access to clean water is mostly considered among the living standards dimensions following the OPHI index. However, it has also been included in the health dimensions due to data constraint on other health indicators and given that it is also a health risk factor [7]. We use the water and sanitation indicators to measure the health dimension for three reasons. First, our analysis is based on *E. coli* contamination test results of the water sample. This measure better captures a health risk than a general living condition. Second, the child mortality data, which is one of the two indicators traditionally included in the health dimension, is only for a one-year period which would understate deprivation in the health dimension. Third, the analysis allows us to compare our results with previous studies based on data collected from the same households [15]. This will allow us to clearly demonstrate the measurement effect of an indicator on MDP levels. The model for this approach is presented in Fig 3. Each indicator in the health dimension is equally weighted and is given a weight of 1/9.

**Table 1. Multidimensional poverty index deprivation indicators and definitions.**

| Deprivation Indicator | Deprived if... |
|---|---|
| Drinking water | *Drinking Water 1*: The household has no access to improved drinking water. |
| | *Drinking Water 2*: The household has no access to safely managed drinking water. |
| | *Drinking Water 3*: The household has no access to safely managed drinking water AND water at point of use free from contamination |
| Sanitation | The household's sanitation facility is not improved according to SDG guidelines). |
| Nutrition | At least one 6–59–month-old child in the household is stunted. |
| Child Mortality | At least one child has died in the household in the 12 months before the survey. |
| Years of schooling | No household member aged 13 or older has completed 6 years of schooling. |
| Child school attendance | At least one child in the household aged 8–15 years is not attending school. |
| Cooking fuel | The household cooks with dung, agricultural crop, shrubs, wood, charcoal, or coal. |
| Electricity | The household has no electricity. |
| Housing | The household has inadequate housing: the floor is made of natural materials or the roof or walls are made of rudimentary materials. |
| Assets | The household does not have an asset that allows access to information (radio, TV, or phone) OR no mobility/transportation asset (animal cart, bicycle, motorbike) AND no livelihood asset (refrigerator, agricultural land, livestock). |

# 3. Results

## 3.1 MPI results from different water quality indicators

Table 2 presents MDP headcount estimates based on the model presented in Fig 3. Additional results are presented in the supporting information tables (S1 and S2 Tables). Nationally, 81 percent of people are multidimensionally poor when safely managed drinking water is used to measure the water indicator. There is about a 12 percentage points (pp) difference between the safely managed indicator and the improved sources indicator which resulted in an MDP of 68.8 percent nationally.

Table 2 also reports the results disaggregated by place of residence. MDP is far worse in rural than in urban areas. Depending on the metric in Table 2, there is about 54 to 60 pp difference in MDP levels between rural and medium and large town areas. In general, MDP decreases as urbanity increases because the most of the MPI indicators are based on amenities that are more available in urban than in rural areas. When the baseline water quality indicator was replaced by the safely managed indicator, the rural-urban difference narrowed slightly because the new indicator increased MDP proportionally more in urban than in rural areas. The change in measurement resulted in an increase in poverty rates in small town areas by 19 pp and in medium and large town areas by 15 pp. The marginal change in rural areas is the smallest because rural households mostly get their drinking water from unimproved sources that are also more likely to be contaminated as illustrated in Fig 2.

MDP headcounts vary from one region to another (Table 3). It is the lowest in Addis Ababa at less than 10 percent. In all the other regions, poverty rates were about 80 percent and more. MDP is highest in the Amhara region on all three scenarios. The change in the water quality measurement indicator resulted in increases in the MDP levels in all regions by about 4–15 pp both at source and at point of use.

Notably the difference between the two safely managed indicators was small compared with the difference between these indicators and no improved source (Tables 2 and 3). The source-point of use difference is more pronounced by area than by region (Table 2). The impact of adjusting for water quality at the point of use was larger for urban areas.

| Health (1/3) | | | Education (1/3) | | Living Standards (1/3) | | | |
|---|---|---|---|---|---|---|---|---|
| Nutrition (1/9) | Drinking water (1/9) | Sanitation (1/9) | Years of schooling (1/6) | Child school attendance (1/6) | Cooking fuel (1/12) | Electricity (1/12) | Housing (1/12) | Assets (1/12) |

**Fig 3. MPI framework with drinking water as a health dimension indicator.** Values in parentheses are weights (contributions to the overall MPI).

### 3.2 Sensitivity analysis

Our main analysis is based on a presence of any level of *E. coli* in the drinking water sample tests. The test, however, comes with different risk levels for different samples. Table 4 reports *E. coli* contamination risk by water source type at both source and point of use using an established scale for the number of *E. coli* detected per 100 mL: low risk (<1), moderate risk (1–10),

**Table 2. Multidimensional poverty headcount and water quality—three indicators for access to drinking water, percent.**

| Area | N | No improved source | | | No safely managed source | | | No safely managed and free from contamination at point of use | | |
|---|---|---|---|---|---|---|---|---|---|---|
| | | H (%) | 95% CI | | H (%) | 95% CI | Diff. (pp) | H (%) | 95% CI | Diff. (pp) |
| National | 4,464 | 68.8 | [65.3–72.3] | | 81.1 | [78.2–84.0] | 12.3 | 82.2 | [79.5–85.0] | 13.4 |
| Rural | 2,993 | 83.7 | [80.6–86.7] | | 94.8 | [92.8–96.8] | 11.1 | 95.2 | [93.2–97.1] | 11.5 |
| Small towns | 366 | 32.0 | [23.6–40.3] | | 51.1 | [42.2–60.0] | 19.1 | 55.0 | [47.6–62.5] | 23.0 |
| Medium & Large towns | 1,105 | 23.6 | [15.9–31.3] | | 38.2 | [29.9–46.5] | 14.6 | 41.5 | [33.5–49.4] | 17.9 |

Notes: Authors' calculation based on ESS 2016. N = number of observations; H (%) = Multidimensional poverty headcount in percent; Diff. (pp) = change in percentage points from the baseline (No improved source).

Table 3. Multidimensional poverty headcount and water quality- three indicators for access to drinking water, by region, percent.

| Regions | N | No improved source | | No safely managed source | | | No safely managed source or not free from contamination at point of use | | |
|---|---|---|---|---|---|---|---|---|---|
| | | H (%) | 95% CI | H (%) | 95% CI | Diff. (pp) | H (%) | 95% CI | Diff. (pp) |
| Addis Ababa | 203 | 5.7 | [0.4–11.1] | 9.2 | [2.3–16.1] | 3.5 | 9.6 | [1.9–17.4] | 3.9 |
| Amhara | 906 | 76.7 | [70.6–82.7] | 89.2 | [85.2–93.1] | 12.5 | 89.8 | [86.0–93.7] | 13.1 |
| Oromia | 913 | 71.7 | [65.5–77.8] | 84.0 | [78.9–89.0] | 12.3 | 85.8 | [81.1–90.4] | 14.1 |
| SNNP[a] | 1,028 | 67.7 | [59.0–76.3] | 82.8 | [75.1–90.4] | 15.1 | 83.1 | [75.6–90.7] | 15.4 |
| Tigray | 531 | 65.3 | [54.5–76.0] | 75.8 | [66.2–85.4] | 10.5 | 77.0 | [67.8–86.1] | 11.7 |
| All Others | 883 | 67.7 | [57.7–77.6] | 78.1 | [70.5–85.7] | 10.4 | 80.4 | [73.5–87.3] | 12.7 |

Notes: Authors' calculation based on ESS 2016. N = number of observations; H (%) = Multidimensional poverty headcount in percent; Diff. (pp) = change n percentage points from the baseline (No improved source). [a]Southern Nations, Nationalities, and Peoples' Region.

high risk (11–100) and very high risk (>100 per 100 mL) [22]. Although at least some water samples from all types of sources tested in the survey were found to be contaminated with *E. coli*, the risk differs by source type. For example, improved sources which include piped and protected types are more likely to be at moderate risk category than unimproved sources. As expected, unimproved source types are in the high and very high-risk categories. Moreover, in general, risk is higher at point of use than at source because of further contamination due to transportation and storage. We looked at the sensitivity of our MPI results to changes in *E. coli* risk levels. Therefore, in this approach we restrict the definition of contamination to the detection of high and very high risk levels.

In a first sensitivity analysis we examined restricted measures that considered only high and very high-risk levels which reduced the deprivation with respect to the water dimension (Table 5). The MDP is slightly lower in the restricted models than in the unrestricted models that are based on moderate and above risk levels. Fig 4 shows the results of both restricted and unrestricted models at source and point of use side by side with the baseline indicator.

In a second set of sensitivity analyses we examined changes in the MPI composition and weighting structures (Table 6). Due to data constraints and other reasons indicated earlier the preferred model for this analysis is based on a modified AF model presented in the previous section. We re-run the MDP analysis utilizing Alkire and Foster's indicators specifications and weights. All the 10 indicators listed in Table 1 are used and the water and sanitation indicators

Table 4. *E. coli* risk levels in drinking water at source and point of use, percent.

| Area | | Low risk (E. coli < 1 cfu/100 mL) | | Moderate risk (E. coli 1–10 cfu/100 mL) | | High risk (E. coli 11–100 cfu/100 mL) | | Very high risk (E. coli >100 cfu/100 mL) | |
|---|---|---|---|---|---|---|---|---|---|
| | N | % | 95%CI | % | 95%CI | % | 95%CI | % | 95%CI |
| *At source* | | | | | | | | | |
| Improved | 3,064 | 22.1 | [20.2–24.1] | 32 | [29.6–34.5] | 26.5 | [24.3–28.7] | 19.5 | [17.6–21.5] |
| Unimproved | 1,462 | 2.4 | [1.5–3.7] | 7.2 | [5.5–9.3] | 22.4 | [19.7–25.4] | 68.0 | [64.7–71.2] |
| All sources | 4,526 | 15.6 | [14.2–17] | 23.8 | [22.1–25.7] | 25.1 | [23.4–26.9] | 35.4 | [33.5–37.4] |
| *At point of use* | | | | | | | | | |
| Improved | 3,201 | 9.8 | [8.7–11.1] | 14.4 | [12.9–16.2] | 43.5 | [41–45.9] | 32.3 | [30–34.6] |
| Unimproved | 1,481 | 0.6 | [0.3–1.2] | 3.9 | [2.7–5.6] | 23.4 | [20.6–26.5] | 72.1 | [68.9–75.2] |
| All sources | 4,682 | 6.9 | [6.1–7.7] | 11.1 | [9.9–12.4] | 37.0 | [35.1–39] | 45.1 | [43.1–47.1] |

Notes: Authors' calculation based on ESS 2016. N = Number of observations.

**Table 5. Multidimensional headcount poverty and water quality–three indicators for access to drinking water, percent.**

| Area | N | No improved source | | High risk level *E. coli* detected at source | | | High risk level *E. coli* detected at point of use | | |
|---|---|---|---|---|---|---|---|---|---|
| | | H (%) | 95% CI | H (%) | 95% CI | Diff. (pp) | H (%) | 95% CI | Diff. (pp) |
| National | 4,464 | 68.8 | [65.3–72.3] | 75.6 | [72.4–78.9] | 6.8 | 79.7 | [76.8–82.4] | 10.9 |
| Rural | 2,993 | 83.7 | [80.6–86.7] | 90.5 | [88.0–93.0] | 6.8 | 93.6 | [91.5–95.7] | 9.9 |
| Small towns | 366 | 32.0 | [23.6–40.3] | 41.4 | [33.2–49.5] | 9.4 | 47.6 | [41.0–54.2] | 15.6 |
| Medium & Large towns | 1,105 | 23.6 | [15.9–31.3] | 29.7 | [2.6–37.7] | 6.1 | 36.9 | [29.0–4.7] | 13.3 |

Notes: Authors' calculation based on ESS 2016. N = Number of observations; H (%) = M multidimensional poverty headcount in percent; Diff. (pp) = Change in percentage points from the baseline (No improved source).

are included in the Living Standards dimension each with a weight of 1/18. However, the findings still hold when the standard AF model is used to calculate the MPI.

The change in the model from modified to the original AF model reduced the change in measurement effect from about 12 pp to about 5pp at the national level (Tables 2 and 6). This

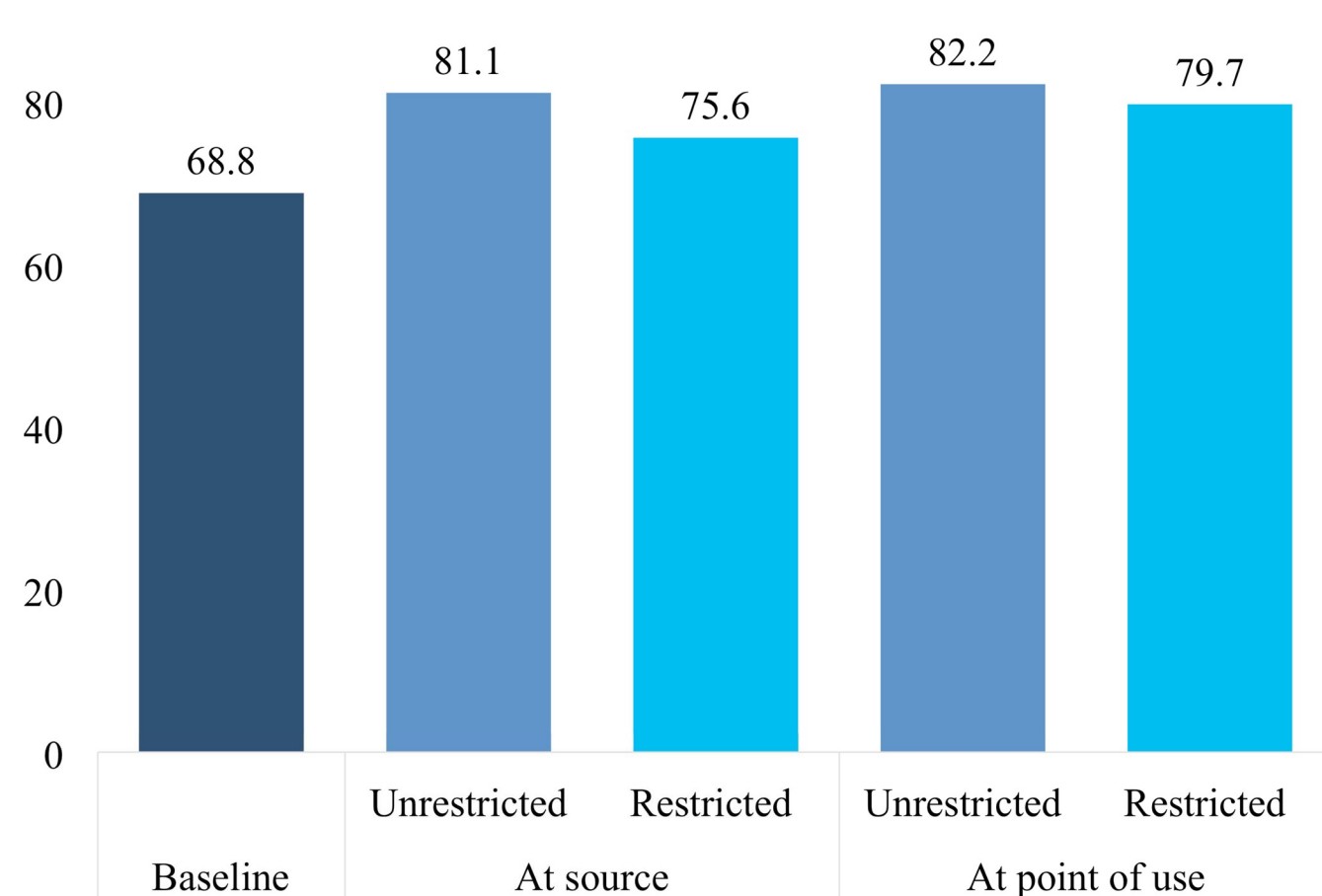

**Fig 4. National level multidimensional headcount poverty with different indicators for access to drinking water, percent.** The unrestricted values are based on safely managed drinking water (Table 2) and the restricted values are based on high and very high risk *E. coli* (Table 5).

**Table 6. Multidimensional headcount poverty and water quality–three indicators for access to drinking water (AF Model), percent.**

| Area | N | No improved source | | No safely managed source | | | No safely managed and free from contamination at point of use | | |
|---|---|---|---|---|---|---|---|---|---|
| | | H (%) | 95% CI | H (%) | 95% CI | Diff. (pp) | H (%) | 95% CI | Diff. (pp) |
| National | 4,464 | 56.3 | [52.6–60.1] | 61.2 | [57.6–64.7] | 4.9 | 61.6 | [58.0–65.1] | 5.3 |
| Rural | 2,993 | 70.0 | [67.4–74.3] | 74.6 | [71.2–77.9] | 4.6 | 74.6 | [71.3–78.0] | 4.6 |
| Small towns | 366 | 23.4 | [16.2–30.5] | 31.8 | [24.2–39.4] | 8.4 | 32.7 | [25.0–40.5] | 9.3 |
| Medium & Large towns | 1,105 | 14.4 | [10.1–18.7] | 19.3 | [14.4–24.2] | 4.9 | 20.7 | [15.9–25.5] | 6.3 |

Notes: Authors' calculation based on ESS 2016. N = Number of observations; H (%) = Multidimensional poverty headcount in percent; Diff. (pp) = Change in percentage points from the baseline (No improved source).

is due to the differences in the weights given to the water indicator in the original and modified models. However, there is still a substantial effect of the change in measurement on MDP due to adjustment to the drinking water indicator.

## 4. Discussion

The Multidimensional Poverty Index (MPI) developed by the Oxford Poverty and Human Development Initiative (OPHI) is widely used to monitor dimensional poverty at global and national levels. However, its effectiveness depends on the quality of the indicators that measure the dimensions. The objective of this study was thus to investigate the effect of changes in measurement of the MPI indicators and the implications for measuring and monitoring welfare. Using data from a recent water quality survey in Ethiopia, we looked at three different indicators of water quality and applied them to estimate MPI at national and regional levels and for rural and urban areas. In our case, depending on the model and scope, replacing the subjective water indicator with a more objective measure raises national MDP by 5–13 pp, which based on the current population estimates is equivalent to 5–13 million more multidimensionally poor. The subjective approach thus understates poverty across groups and regions. The finding is robust to changes in MPI weighting and aggregation structure and contamination risk levels.

The results suggest the following messages for data collection and dimensional poverty analysis:

1. Indicators based on self-reports or subjective measurement approaches can provide incomplete information about a poverty dimension. In the case of drinking water quality the direct measurement not only allows to measure quality of the water that the household has access to but also the quality of drinking water actually consumed. We find a substantial difference in poverty levels when comparing self-reported use of improved sources of drinking water with indicators that incorporate direct measurements of drinking water quality.

2. The bias from self-report or subjective measurement has differential effects on the poverty levels of different groups. Although MPI increased for all groups in both rural and urban areas, it increased more for urban than for rural dwellers. The same holds for regional differences. The change in some regions is proportionally higher than in others with important implications for targeting of policy and programmatic interventions to address multidimensional poverty.

3. With the new focus on water quality as part of the indicator for SDG target 6.1 "use of safely managed drinking water services", data on water quality have been collected in household

surveys from an increasing number of low- and middle-income countries. Globally the proportion of the population using *unimproved* sources has been estimated by the WHO/UNICEF Joint Monitoring Programme as 8 percent (578 million) whereas around 29 percent (2.1 billion) used drinking water from either an unimproved source or a source that is contaminated [11]. Household surveys in several other countries in sub-Saharan Africa have also found that drinking water is often contaminated at the source, with the proportion of the population with drinking water with *E. coli* detected ranging from 43 percent in Ghana to 77 percent in Nigeria and 90 percent in Sierra Leone [12]. This analysis could be adapted for these and other countries that have integrated water testing in household surveys, with the increases in MDP expected to varying depending on the difference between improved and free from contamination as well as the proportion of the population close to the MDP threshold.

4. While data availability limited our analysis to a single indicator, other MPI indicators are known to have their own measurement problems. In education, literacy and numeracy test results are better measures of learning outcomes than enrolment and grade completion [8]. Similarly, indicators for access to cooking fuel and electricity need to include safety and sufficiency aspects to measure energy poverty [9]. The use of additional indicators aligned with the ambition of other SDG targets, such as SDG 6.2 (safely managed sanitation) and SDG 1.4 (basic services), would further increase multidimensional poverty provided thresholds for the MPI are not adjusted. The impact of improving the measurement of multiple indicators simultaneously is likely to increase the level of poverty more than adjusting one measurement at a time, as we have done in this study. More broadly, the SDG targets and indicators are considerably more ambitious than those originally used to define the MPI's dimensions and a comprehensive assessment aligned to the SDGs would be expected to greatly increase multidimensional poverty.

The study has several limitations. First, the analysis is based on a one-off assessment of drinking water quality and is unlikely to reflect year-round safety. The study was conducted during the wet season in Ethiopia (May-July) and as a result contamination may have been more frequently detected than had the study taken place during the dry season [23]. The impacts of seasonality on different faecal contamination pathways may differentially impact different types of water supply and it cannot be assumed that contamination is worst in the wet season as illustrated in a recent study in Uganda [24]. The cross-sectional nature of the survey data mean we are unable to examine the impact of adjusting for water quality in other seasons. The adjusted MPI may therefore be sensitive to the timing of water quality testing, and affected by dry-season bias in household surveys [25]. Second, the study addressed only *E. coli* contamination. However, water quality can be affected by numerous microbial and chemical agents as outlined in the WHO Guidelines for Drinking-Water Quality [26]. For example, excessive fluoride is also a known issue in some parts of the country but we did not adjust for this parameter given the comparatively low proportion of the population (3.8% exceeded >1.5 mg/L) affected according to laboratory results from the ESS [12]. Future studies could examine a wider range of health-based water quality parameters. In addition, our study explored MPI because it is a widely used index of monitoring dimensional poverty. However, there are other dimensional indices that are used to assess poverty and human development. The adjustment for water quality is expected to affect these indices to varying degrees and the extent of the adjustment would also be affected by other changes to indices that seek to align with SDG targets for other dimensions including health, education and energy.

## 5. Conclusion

MPI is an important and widely used approach to monitor poverty in developing countries. Its effectiveness depends on identifying the right indicators and measuring them correctly. Using a recent survey on water quality in Ethiopia, we examined the effect of accounting for drinking water quality on multidimensional poverty. We compared the traditional measure for drinking water access ("improved source") with the new SDG indicator ("safely managed") which incorporates direct measurement of water quality, using an indicator for faecal contamination. Our study demonstrates that accounting for water quality has a substantial impact on the poverty headcount in Ethiopia and that reliance on reported use of different types of water source as the measure for drinking water access can greatly understate poverty. The effect on poverty levels varies between regions and is highest in urban Ethiopia where coverage of improved sources is highest. Although an increasing number of countries have implemented water quality testing in household surveys, current approaches can be costly and logistically complex for national statistical offices and it is unlikely that all surveys will include this module. A stricter definition of the deprivation indicators that are predicted based on existing self-reported data or based on integration of water quality data from administrative datasets could potentially minimize the bias.

## Supporting information

**S1 Fig. Improved and unimproved water source types and their *E. coli* contamination status at source and point of use in rural areas.**
(DOCX)

**S2 Fig. Improved and unimproved water source types and their *E. coli* contamination status at source and point of use in urban areas.**
(DOCX)

**S1 Table. Summary of MPI indicator for three scenarios.**
(DOCX)

**S2 Table. MPI results for three scenarios by region and residence type.**
(DOCX)

## Acknowledgments

The authors would like to thank anonymous referees for valuable comments and suggestions.
Disclaimer
The findings, interpretations, and conclusions expressed in this paper are entirely those of the authors. They do not necessarily represent the views of UNICEF or the World Bank and its affiliated organizations, or those of the Executive Directors of UNICEF or the World Bank or the governments they represent.

## Author Contributions

**Conceptualization:** Alemayehu Azeze Ambel, Harriet Kasidi Mugera, Robert E. S. Bain.

**Formal analysis:** Alemayehu Azeze Ambel, Harriet Kasidi Mugera.

**Methodology:** Alemayehu Azeze Ambel, Harriet Kasidi Mugera, Robert E. S. Bain.

**Writing – original draft:** Alemayehu Azeze Ambel, Harriet Kasidi Mugera.

**Writing – review & editing:** Alemayehu Azeze Ambel, Harriet Kasidi Mugera, Robert E. S. Bain.

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
