## [Decision Letter · Decision Letter 0]

21 Aug 2020

PONE-D-20-15278

Accounting for Drinking Water Quality in Measuring Multidimensional Poverty in Ethiopia

PLOS ONE

Dear Dr. Ambel,

Thank you for submitting your manuscript to PLOS ONE. After careful consideration, we feel that it has merit but does not fully meet PLOS ONE’s publication criteria as it currently stands. Therefore, we invite you to submit a revised version of the manuscript that addresses the points raised during the review process.

Three reviewers consistently point to issues regarding the proper justification and measurement of the proposed Multidimensional Poverty Index. All of their observations should be addressed in a revised submission.

We look forward to receiving your revised manuscript.

Kind regards,

Francisco X Aguilar

Academic Editor

PLOS ONE

Journal Requirements:

'AA and HKM work for the World Bank'

Reviewers' comments:

Reviewer's Responses to Questions

**Comments to the Author**

1. Is the manuscript technically sound, and do the data support the conclusions?

Reviewer #1: Yes

Reviewer #2: Yes

Reviewer #3: Yes

2. Has the statistical analysis been performed appropriately and rigorously? 

Reviewer #1: Yes

Reviewer #2: Yes

Reviewer #3: Yes

3. Have the authors made all data underlying the findings in their manuscript fully available?

Reviewer #1: Yes

Reviewer #2: Yes

Reviewer #3: Yes

4. Is the manuscript presented in an intelligible fashion and written in standard English?

Reviewer #1: Yes

Reviewer #2: Yes

Reviewer #3: Yes

5. Review Comments to the Author

Reviewer #1: This study clearly shows that accounting for drinking water quality in measuring multidimensional poverty increases the headcount of the latter. The paper is focused and easy to follow.

The effect of accounting for drinking water quality on multidimensional poverty has qualitative and quantitative aspects. Qualitatively, using a more precise measure of an indicator (safe water in this case) is likely to make headcount poverty always higher. Reading this paper doesn’t add much insight in terms of the qualitative effect and hence this aspect doesn’t justify the paper well. That is, the conclusion that “reliance on reported use of different types of water source as the measure of drinking water access can greatly understate poverty” is all too obvious, even without doing such a study.

The quantitative measure of the effect (that poverty increases 5-13 pp), on the other hand, may suffer from several limitations, some mentioned by the author(s) and others in my comments below.

One, data were collected May-July, the wet months in most parts of Ethiopia. E.coli contamination is likely to be much higher during these months, since even relatively safer sources will be flooded. This is likely to make the figures unreliable, as having a survey a months earlier or later might affect the results.

Two, the sampling units are households but safety of drinking water at source is the same for people who fetch water from the same river. This creates clusters of households which will have similar measures of water quality. Another way of thinking about this is that sampling over households and sampling over sources will not be the same.

Three, the claim that “all [households and sources] are statistically representative nationally” is too strong, considering the fact that water quality can be sensitive to changes such as rain and flooding, topography, culture and so forth.

Other comments:

A small section describing poverty, access to water, rural-urban differences and natural environmental differences would give context to the results.

Page 4, “It is also an effective tool for targeting.” Not all potential readers of this paper will understand ‘targeting’ the same way the authors think about it.

Page 7, “The percentages are weighted.” It may not be clear how and why.

Table 1,

Child Mortality: children dying of other natural causes included.

Years of schooling: no 10 year-old is old enough to have completed 6 years of schooling.

Cooking fuel: is a household that uses charcoal to make coffee included? This is important, given the cultural significance of making coffee.

Reviewer #2: This paper investigates the accuracy of the Multidimensional Poverty Index (MPI) by replacing one of the measurements in the index (water source type) to a more comprehensive measure (safely managed drinking water services, defined as water free from E. coli contamination). By using this new measure of MultiDimensional Poverty (MPD) on data from Ethiopia the authors find that the Multidimensional Poverty Headcount (MPH) increase by 13 percentage points on the national level, or at least by 5 percentage points using a measure of E. coli risk. This can be translated to between 5-13 million more people in MDP.

The paper does not use any advanced statistical methods, but asks a relatively straightforward question: do we measure MPD correctly?

I share the authors conviction that it is important with good measurements on such an important topic as poverty, but I do lack a discussion in the paper on the take-home message from the paper. While I have no doubt that water quality is very important – especially for the very poor – I am not necessarily convinced that we should redefine the MPI, which is the implicit message I read in the paper. We can define “poverty” how we like; the relevant question to ask of any measurement is that of validity and reliability. An even more strict definition of (good) water quality than the authors use would no doubt increase the MPH in Ethiopia further. Having access to improved water sources (even if not necessarily “clean”), is likely to be better than not having access to it at all. Maybe this is the correct definition? If not, why?

Thus, the fundamental question I ask is: Does it matter? I do think that the paper would be much improved if it gives a more thorough discussion on why this new MPI definition is to be preferred. Are there any policy recommendations etc.?

The authors should clarify the discussion on subjective or objective measure of water, as this is important (which is clear from the discussion and conclusion). On page 15 (line 298-299) the authors write “replacing the subjective water indicator with a more objective measure…”. On page 5 (line 91) the reader gets the impression that the standard way of measuring is objective, but possibly not a good measure as it only focus on whether the water source is improved or not.

* The tables should present the number of observations.

* The share of the population living in different areas (urban, rural etc.) should be presented.

Reviewer #3: This paper estimates multidimensional headcount poverty in Ethiopia. The authors find that using a direct measure of water quality (whether E. coli is present at the water source of the household) instead of the more commonly used survey measure of access to drinking water (whether the household has access to an improved water source) increases multidimensional headcount poverty by at least 5 percentage points.

This paper shows the importance of measurement. Thus, two revisions could improve this paper:

1. The authors should be clearer about the exact measures related to drinking water. The authors refer to a safely managed water source, which I believe is measured as no E. coli present at the water source of the household. The authors should explicitly state this so that the text reads more clearly. For the measure of access to drinking water, defined as whether the household has access to an improved water source, the authors refer to this measure as self-reported. Who reports this measure and how is the question asked?

2. The implication of the two measures is that the direct measure of the presence of E. coli is accurate while the self-reported measure of access to an improved water source is measured with error. To support this implication, how accurate are the direct measurements in reflecting the presence of E. coli throughout the year? Is E. coli the most harmful contaminant in the drinking water in Ethiopia? Could the self-reported measure of access to an improved water source reflect water quality based on other contaminants besides E. coli? The authors address the limitations briefly in the last paragraph of the Discussion section. However, since measurement is the important theme of the paper, these limitations should be further discussed. Evidence from other studies could be used to describe whether these limitations are likely to be minor and not change the main conclusions from this paper.

6. PLOS authors have the option to publish the peer review history of their article (what does this mean?). If published, this will include your full peer review and any attached files.

Reviewer #1: No

Reviewer #2: No

Reviewer #3: No

---

## [Author Response · Author response to Decision Letter 0]

21 Oct 2020

Response to reviewer comments

Reviewer #1: This study clearly shows that accounting for drinking water quality in measuring multidimensional poverty increases the headcount of the latter. The paper is focused and easy to follow.

Response: Many thanks for your review and positive comments regarding the focus and clarity of the study. 

The effect of accounting for drinking water quality on multidimensional poverty has qualitative and quantitative aspects. Qualitatively, using a more precise measure of an indicator (safe water in this case) is likely to make headcount poverty always higher. Reading this paper doesn’t add much insight in terms of the qualitative effect and hence this aspect doesn’t justify the paper well. That is, the conclusion that “reliance on reported use of different types of water source as the measure of drinking water access can greatly understate poverty” is all too obvious, even without doing such a study.

Response: We agree the direction of the effect is somewhat obvious/predictable and that the magnitude of the adjustment is a key aspect of this study. Notably we find differential impacts of the adjustment for different population groups in Ethiopia. Water quality testing in household surveys is more logistically complex than simply asking households what type of water source they use and it is entirely reasonable to question whether this new measure has a substantial impact on assessments of “access”. In this study we sought to address whether the adjustment had a quantitative impact on poverty headcounts as this is a critical indicator for Governments and other policy makers allocating resources. 

The quantitative measure of the effect (that poverty increases 5-13 pp), on the other hand, may suffer from several limitations, some mentioned by the author(s) and others in my comments below.

Response: We acknowledge the limitations of an assessment of quality based on a “snapshot” of water quality offered by integrating water quality testing in a national survey. We have expanded our discussion of the limitations of a focus on E. coli only and the cross-sectional nature of the survey data. 

One, data were collected May-July, the wet months in most parts of Ethiopia. E.coli contamination is likely to be much higher during these months, since even relatively safer sources will be flooded. This is likely to make the figures unreliable, as having a survey a months earlier or later might affect the results.

Response: The wet months are indeed likely to be a time when water sources are more often contaminated with FIB. We have expanded on seasonality of water quality and household surveys in the limitations section.

Two, the sampling units are households but safety of drinking water at source is the same for people who fetch water from the same river. This creates clusters of households which will have similar measures of water quality. Another way of thinking about this is that sampling over households and sampling over sources will not be the same.

Response: The ESS-WQ sample is based on a multi-stage cluster sample of households in Ethiopia and the sources used by these households. This means that all of the data in this study relate to the % of the population rather than the % of water sources. We consider this an advantage rather than a disadvantage of water quality testing in household surveys – firstly because a reliable sampling frame is available for households but not for water sources and because we are interested in population access to services as an indicator of deprivation following the MPI methodology. Note that similar considerations apply for the standard MPI improved source indicator and also to other dimensions of the MPI, for which intra-cluster correlation can be high. Furthermore, the study examined quality at point of use which yielded similar conclusions to the adjustment based on source water quality. 

Three, the claim that “all [households and sources] are statistically representative nationally” is too strong, considering the fact that water quality can be sensitive to changes such as rain and flooding, topography, culture and so forth.

Response: We have expanded our discussion of the limitations of a cross-sectional assessment of water quality (as above) and have clarified this statement as follows: “the overall sample is representative of households in Ethiopia and for six major regions”. The household survey sample is statistically representative and drawn from a robust sampling frame. 

Other comments:

A small section describing poverty, access to water, rural-urban differences and natural environmental differences would give context to the results.

Response: We have added a new paragraph describing access to drinking water services in Ethiopia including between rural and urban areas. 

Page 4, “It is also an effective tool for targeting.” Not all potential readers of this paper will understand ‘targeting’ the same way the authors think about it. Response: We have clarified meaning of targeting “… identifying priority areas for programmatic and policy interventions. “

Page 7, “The percentages are weighted.” It may not be clear how and why. Response: This section of the methods has been revised. We have used weights throughout the analysis as explained in the methods. 

Table 1,

Child Mortality: children dying of other natural causes included. 

Years of schooling: no 10 year-old is old enough to have completed 6 years of schooling. 

Cooking fuel: is a household that uses charcoal to make coffee included? This is important, given the cultural significance of making coffee. 

Response: Overall, other indicators are included following the standard MPI aggregation and definitions. We are taking all other indicators as they are and changing only the water indicator. On child mortality, this is all cause mortality. On cooking fuel, the deprivation indicator is based on the main source of cooking fuel. Households can still use charcoal and others for coffee and even some cooking in the absence of the main source of cooking fuel. On years of schooling, thank you for highlighting the mistake related to 10 year-olds and above which we have now corrected (13 and above).

 

Reviewer #2: This paper investigates the accuracy of the Multidimensional Poverty Index (MPI) by replacing one of the measurements in the index (water source type) to a more comprehensive measure (safely managed drinking water services, defined as water free from E. coli contamination). By using this new measure of MultiDimensional Poverty (MPD) on data from Ethiopia the authors find that the Multidimensional Poverty Headcount (MPH) increase by 13 percentage points on the national level, or at least by 5 percentage points using a measure of E. coli risk. This can be translated to between 5-13 million more people in MDP.

The paper does not use any advanced statistical methods, but asks a relatively straightforward question: do we measure MPD correctly?

I share the authors conviction that it is important with good measurements on such an important topic as poverty, but I do lack a discussion in the paper on the take-home message from the paper. While I have no doubt that water quality is very important – especially for the very poor – I am not necessarily convinced that we should redefine the MPI, which is the implicit message I read in the paper. We can define “poverty” how we like; the relevant question to ask of any measurement is that of validity and reliability. An even more strict definition of (good) water quality than the authors use would no doubt increase the MPH in Ethiopia further. Having access to improved water sources (even if not necessarily “clean”), is likely to be better than not having access to it at all. Maybe this is the correct definition? If not, why?

Thus, the fundamental question I ask is: Does it matter? I do think that the paper would be much improved if it gives a more thorough discussion on why this new MPI definition is to be preferred. Are there any policy recommendations etc.?

Response: The MPI provides a framework for assessment of non-monetary poverty and is widely used. In this study we have examined the impact of adjusting the index to adapt it for the SDG for drinking water services. This also relates to the Human Right to Water which places emphasis on the availability, accessibility and quality of drinking water.

We agree that accounting for water quality is important, and especially affects the poorest households. The purpose of this study was not to universally recommend a change to the MPI definition but rather to explore the impact of substituting the standard indicator with an indicator aligned with the new global benchmark for drinking water services. We acknowledge in the discussion that this might “unbalance” the dimensions, and that similar adjustments would be needed to meet the SDG standards for health, education etc. In response to the reviewer’s comment we have placed greater emphasis on the policy implications in the discussion.

The authors should clarify the discussion on subjective or objective measure of water, as this is important (which is clear from the discussion and conclusion). On page 15 (line 298-299) the authors write “replacing the subjective water indicator with a more objective measure…”. On page 5 (line 91) the reader gets the impression that the standard way of measuring is objective, but possibly not a good measure as it only focus on whether the water source is improved or not.

Response: Many thanks for raising this. We have clarified the distinction as “self-reported” (use of improved sources) and “direct measure” (free from contamination). We have also clarified the definitions used for the drinking water dimension in response to reviewer #3s comments.

* The tables should present the number of observations. 

Response: We have added number of observations in all our tables.

* The share of the population living in different areas (urban, rural etc.) should be presented. 

Response: We have added the share of the total population living in different areas as supplementary materials. 

 

Reviewer #3: This paper estimates multidimensional headcount poverty in Ethiopia. The authors find that using a direct measure of water quality (whether E. coli is present at the water source of the household) instead of the more commonly used survey measure of access to drinking water (whether the household has access to an improved water source) increases multidimensional headcount poverty by at least 5 percentage points.

This paper shows the importance of measurement. Thus, two revisions could improve this paper:

1. The authors should be clearer about the exact measures related to drinking water. The authors refer to a safely managed water source, which I believe is measured as no E. coli present at the water source of the household. The authors should explicitly state this so that the text reads more clearly. For the measure of access to drinking water, defined as whether the household has access to an improved water source, the authors refer to this measure as self-reported. Who reports this measure and how is the question asked?

Response: We have clarified definitions of drinking water access in the methods section and explained how the question on drinking water sources were formulated:

• Drinking water indicator 1: No improved source. This measure depends on the self-reported response that was provided by the household on the type of water source used for drinking. The head of the household or any other most eligible member of the household was the respondent to these questions as they are believed to be the most informed members of the household on water related issues in the household. Water sources were classified as improved or unimproved according to the definitions of the WHO/UNICEF JMP.

• Drinking water indicator 2: No safely managed source. This measure combines self-reported information on the type of water source, location and availability with direct measurement of water quality. Households were considered to lack a safely managed drinking water source if they used an unimproved drinking water source or relied on an improved source of drinking water which was located off premises, did not provide sufficient drinking water in the preceding month, or from which a measure of fecal contamination (≥1 E. coli in a 100 ml sample) was detected in water quality samples collected from the source. This corresponds to the new SDG benchmark for drinking water services as monitored by WHO/UNICEF JMP. 

• Drinking water indicator 3: No safely managed source free from contamination at point of use. As above but also considering water quality at the point of use. In addition to the criteria for safely managed drinking water services, households were considered to be deprived if E. coli were detected in the sample of drinking water at the point of use. This is a more stringent standard than used by WHO/UNICEF JMP as it reflects the quality of drinking water at the point of consumption.

We have also included the exact formulation of the questions used to define these indicators in supplementary materials. 

2. The implication of the two measures is that the direct measure of the presence of E. coli is accurate while the self-reported measure of access to an improved water source is measured with error. To support this implication, how accurate are the direct measurements in reflecting the presence of E. coli throughout the year? Is E. coli the most harmful contaminant in the drinking water in Ethiopia? Could the self-reported measure of access to an improved water source reflect water quality based on other contaminants besides E. coli? The authors address the limitations briefly in the last paragraph of the Discussion section. However, since measurement is the important theme of the paper, these limitations should be further discussed. Evidence from other studies could be used to describe whether these limitations are likely to be minor and not change the main conclusions from this paper.

Response: We have elaborated on limitations of E. coli, exclusive focus on microbial contamination and seasonality of this indicator (see earlier responses). Note fluoride was also included in ESS but found to impact a much smaller proportion of the population, concentrated in the rift valley.

---

## [Decision Letter · Decision Letter 1]

1 Dec 2020

Accounting for Drinking Water Quality in Measuring Multidimensional Poverty in Ethiopia

PONE-D-20-15278R1

Dear Dr. Ambel,

We’re pleased to inform you that your manuscript has been judged scientifically suitable for publication and will be formally accepted for publication once it meets all outstanding technical requirements.

Kind regards,

Francisco X Aguilar

Academic Editor

PLOS ONE

Additional Editor Comments: Authors shall address minor issues raised during the last iteration.

Reviewers' comments:

Reviewer's Responses to Questions

**Comments to the Author**

1. If the authors have adequately addressed your comments raised in a previous round of review and you feel that this manuscript is now acceptable for publication, you may indicate that here to bypass the “Comments to the Author” section, enter your conflict of interest statement in the “Confidential to Editor” section, and submit your "Accept" recommendation.

Reviewer #2: All comments have been addressed

Reviewer #3: All comments have been addressed

2. Is the manuscript technically sound, and do the data support the conclusions?

Reviewer #2: Yes

Reviewer #3: Yes

3. Has the statistical analysis been performed appropriately and rigorously? 

Reviewer #2: Yes

Reviewer #3: Yes

4. Have the authors made all data underlying the findings in their manuscript fully available?

Reviewer #2: Yes

Reviewer #3: Yes

5. Is the manuscript presented in an intelligible fashion and written in standard English?

Reviewer #2: Yes

Reviewer #3: Yes

6. Review Comments to the Author

Reviewer #2: I thank the authors for their reply and revised manuscript.

I have a minor point that I noticed during the re-reading of the manuscript. In Table 2 and Table 6, I do believe that it should be an "or" instead of an "and" in the last column header. I.e., "No safely managed OR free from contamination at point of use". As it stands now, the numbers makes no sense as an "and" is more strict than an "or", hence "H (%)" should be lower -- not higher. Compare with Table 3. To avoid confusion, it seems best to use the column header from Table 3 in Table 2 and 6.

Reviewer #3: (No Response)

7. PLOS authors have the option to publish the peer review history of their article (what does this mean?). If published, this will include your full peer review and any attached files.

Reviewer #2: No

Reviewer #3: No

---

## [Editor Report · Acceptance letter]

4 Dec 2020

PONE-D-20-15278R1 

Accounting for Drinking Water Quality in Measuring Multidimensional Poverty in Ethiopia 

Dear Dr. Ambel:

I'm pleased to inform you that your manuscript has been deemed suitable for publication in PLOS ONE. Congratulations! Your manuscript is now with our production department. 

Kind regards, 

on behalf of

Dr. Francisco X Aguilar 

Academic Editor

PLOS ONE